# Th1/Th2 Immune Imbalance in the Spleen of Mice Induced by Hypobaric Hypoxia Stimulation and Therapeutic Intervention of Astragaloside IV

**DOI:** 10.3390/ijms26062584

**Published:** 2025-03-13

**Authors:** Rong Gao, Zhenhui Wu, Wanyun Dang, Tingyu Yang, Junru Chen, Hongbo Cheng, Jialu Cui, Lin Lin, Xin Shen, Fangyang Li, Jiayi Yan, Yehui Gao, Yue Gao, Zengchun Ma

**Affiliations:** 1School of Pharmacy, Guangdong Pharmaceutical University, Guangzhou 510006, China; gaoronger198@163.com (R.G.); wanyundang@163.com (W.D.); 15088094928@163.com (J.C.); chb0000@163.com (H.C.); lifangyang7012@163.com (F.L.); 2Beijing Institute of Radiation Medicine, Beijing 100859, China; wzh77580@163.com (Z.W.); 15254555178@163.com (T.Y.); cuijialu143021@163.com (J.C.); neimenggu2022@126.com (L.L.); shenxin9204@126.com (X.S.); yjy19951025@163.com (J.Y.); g1076192612@163.com (Y.G.); 3School of Pharmacy, Jiangxi University of Chinese Medicine, Nanchang 330000, China; 4School of Chinese Materia Medica, Tianjin University of Traditional Chinese Medicine, Tianjin 301617, China

**Keywords:** hypobaric hypoxia, spleen, immune imbalance, Th1/Th2, Astragaloside IV

## Abstract

This study aims to establish a hypobaric hypoxia-induced immune injury model and investigate the intervention and therapeutic effects of Astragaloside IV (AS-IV). This study simulated hypobaric hypoxia stimulation in mice at an altitude of 7000 m on a plateau for 1, 3, 5, and 7 days. HE staining and transcriptomic analysis were performed on mouse spleens. In addition, AS-IV was selected for intervention in prevention and treatment, and validated by flow cytometry, ELISA, and Q-PCR. The results showed that under simulated hypoxic conditions at an altitude of 7000 m for 5 days, the peripheral blood lymphocytes of mice decreased, and the CD45^+^ cells, CD3^+^ T cells, and CD3^+^CD4^+^ T cells, and CD4^+^/CD8^+^ cell ratio in the spleen all decreased. AS-IV can significantly alleviate pathological damage to the spleen, decrease serum levels of IL-2 and IL-6, increase IL-4 and IL-10, and raise CD3^+^CD4^+^ T cells and the CD4^+^/CD8^+^ cell ratio in peripheral blood and the spleen, while increasing CD4^+^IFN-γ^+^cells in spleen, reducing ROS and apoptosis levels in spleen, and increasing the content of relevant mRNA in the Th1/Th2 cell pathway. In summary, simulating hypoxia at an altitude of 7000 m for 5 days can establish a stable hypobaric hypoxic immune injury model, and AS-IV can effectively alleviate hypobaric hypoxic immune injury.

## 1. Introduction

The most distinctive feature of high-altitude regions is their extremly hypoxic environment, which significantly impacts the physiological functions of the body [1]. Research has demonstrated that hypoxia exerts fundamental regulatory effects on immune and inflammatory responses, revealing that exposure to hypobaric hypoxia can multidimensionally disrupt immune system homeostasis [2,3]. This disruption encompasses multiple levels, including immune responses, inflammatory cascades, and metabolic pathways [4]. Specifically, under hypoxic conditions, there is a notable shift in the balance of cytokine secretion within the body, leading to an imbalance between pro-inflammatory and anti-inflammatory reactions [5,6,7]. This phenomenon is hypothesized to be closely associated with the imbalance in the proportions of immune cell subpopulations. This study focuses on the dynamic changes in immune cells in hypobaric hypoxia environments to understand this complex physiological process and explore potential intervention strategies. Additionally, it evaluates the potential benefits of pharmacological interventions in restoring immune homeostasis and modulating inflammatory responses.

The immune system plays a critical role in maintaining homeostasis through its functions of immune surveillance, defense, and regulation [8]. Inflammation and hypoxia are often interrelated in many pathological conditions: on one hand, inflammatory responses can lead to localized or systemic hypoxia, known as inflammatory hypoxia; on the other hand, hypoxic environments can exacerbate inflammatory responses by activating inflammation-related pathways and altering immune cell function [9]. Hypoxic conditions result in changes in the function and number of different lymphocytes, with an imbalance in cellular immunity being a significant factor in triggering inflammatory responses. This imbalance is most prominently manifested by the dysregulation of CD4^+^ T lymphocyte subsets [10]. T lymphocytes, as a core component of adaptive immune responses, play a crucial role in combating infectious diseases and cancer [11]. Studies have shown that in a hypobaric hypoxic environment, the total number of CD3^+^ T cells in the body tends to decrease, primarily due to a reduction in CD4^+^ T cells [7]. However, there is relatively limited research on how specific types of helper T cells, such as Th1 and Th2, are affected by hypobaric hypoxia. Therefore, further investigation into the specific patterns and mechanisms of T cell changes under these unique conditions is warranted. Understanding these dynamics will provide valuable insights into the immune response alterations caused by hypoxia and may identify potential therapeutic targets for mitigating hypoxia-induced immune dysfunction.

*Astragalus membranaceus Bunge*, a traditional Chinese medicine, has been used in Traditional Chinese Medicine (TCM) for over 2000 years. It primarily grows in provinces such as Shanxi, Inner Mongolia, and Gansu in China, as well as in Korea and Mongolia [12]. Known for its ability to invigorate qi and promote blood circulation, Astragalus has long been employed by TCM practitioners to enhance human immunity [13]. AS-IV, one of the main active components extracted from Astragalus, exhibits a wide range of pharmacological effects, including but not limited to immune modulation, anti-inflammatory, antioxidant, and metabolic regulation properties [14]. Studies have shown that AS-IV can be used to treat various immune-related diseases with minimal toxicity and side effects, modulating immune function by influencing the production of immunologically active substances [15]. Specifically, AS-IV regulates the activities of multiple types of immune cells, such as macrophages, natural killer cells, and lymphocytes, promoting the generation of important cytokines like IL-2 and IFN-γ [16]. This leads to improved function of key immune organs, such as the spleen, thereby achieving an immune-regulatory effect [17]. Additionally, AS-IV has been shown to significantly impact CD4^+^ T cell subsets, including Th1 and Th2 cells, contributing to reduced inflammatory responses [18]. Despite these findings, there is currently a lack of studies investigating the specific effects of AS-IV on immune system damage caused by hypobaric environments. Therefore, this study aims to explore the influence of AS-IV on inflammation and immune cell changes in mice living at high altitudes. Through this research, we hope to uncover the potential value of AS-IV in alleviating discomfort associated with hypobaric exposure, providing theoretical support for the development of new therapeutic approaches.

## 2. Results

### 2.1. Establishment of Hypobaric Hypoxia Immune Injury Model

#### 2.1.1. Effects of Different Times of Hypobaric Hypoxia on the Physiological Indices in Mice

The schematic diagram of the experimental protocol is shown in Figure 1A. There were no deaths in the experimental and control groups of mice until the time of sacrifice. Body weight is a reflection of the general metabolism of the organism, and the changes in body weight are shown in Figure 1B. The body weight of the mice decreased significantly after the treatment with hypobaric hypoxia at an altitude of 7000 m. The symptom of body weight loss appeared to be alleviated after the sixth day, but it was still significantly reduced. The fluctuation of white blood cell (WBC) content and its composition were indicators of changes in the immune response. The changes in WBC content were not statistically significant compared to the blank group (Figure 1C). Lymphocyte percentage (LYMPH%) was decreased, significantly in the H5 group (Figure 1D). Neutrophil percentage (NEUT%) was elevated, with the highest value in the H5 group, with a tendency to recover on day 7 (Figure 1E). Hypobaric hypoxia resulted in a significant increase in RBCs (Appendix A), suggesting compensatory overproliferation of erythrocytes during exposure to hypobaric hypoxia in the plateau; the monocyte percentage (MONO%) was significantly elevated on day 7 (Appendix A). Monocytes are essential innate immune cells that shape the immune response through their role in tissue healing, pathogen clearance and activation of the adaptive immune system [19]. Eosinophil percentage (EO%) was significantly elevated in H3 and H7, with no significant change in H1 and H5 (Appendix A). Basophil percentage (BASO%) was significantly higher in the H7 group (Appendix A). Platelet count (PLT) tended to decrease significantly overall with prolonged hypobaric hypoxia (Appendix A). Hemoglobin (HGB) content increased significantly with the increase in hypobaric hypoxia time (Appendix A). Compared with the NC group, IL-2 and IL-6 were significantly elevated in the serum of mice in the model group (Figure 1F,H), and IL-4 and IL-10 were significantly decreased (Figure 1G,I). This suggests that hypobaric hypoxia leads to elevated serum pro-inflammatory factors and decreased anti-inflammatory factors in mice.

#### 2.1.2. Effects of Different Hypobaric Hypoxia Times on Splenic Index and Pathological Changes of Spleen in Mice

Compared with the control group, mice exposed to hypobaric hypoxia showed leanness, decreased locomotion and shortness of breath (Figure 2A). The spleen of mice in the H5 group appeared to be enlarged. Compared with the blank control group, the spleen coefficients of mice treated with hypobaric hypoxia at an altitude of 7000 m showed a trend of decreasing, then increasing and then decreasing with time (Figure 2B,C). It was suggested that the decrease in the spleen index on the first day of hypoxia was due to the result of mice adapting to the acute plateau hypoxia response, while the significant increase in the spleen index afterward might be related to the stress erythropoiesis [20]. HE staining results showed that the spleen tissue structure of the NC group was basically normal, the splenic tubercle structure was complete in the field of view, the red and white medulla were clearly demarcated, the lymphocytes in the white medulla were closely arranged and abundant, and no necrosis was observed.

Compared with the NC group, the splenic tissue structure of the H1 group was mildly abnormal, with a slight fusion of the red and white medulla, tightly arranged lymphocytes in the white medulla, and a decrease in the number of lymphocytes, as shown by the blue arrowheads, while the lymphocytes in the red medulla were fewer in number, sparsely arranged, and showed a slight inflammatory infiltration, as shown by the black arrowheads. The splenic tissue structure of the H3 and H5 groups was moderately abnormal, with scattered structure of the splenic tubercle in the field of view, blurred boundaries between the red and white medullas, obvious fusion of the red and white medullas, and reduced number of lymphocytes, as shown by the blue arrows; obvious infiltration of inflammatory cells was seen in the tissue as shown by the black arrows. In group H7, the splenic tissue structure was mildly abnormal, the splenic tubercle structure was intact in the field of view, the red and white medullas were clearly demarcated, and the lymphocytes in the white medulla were sparsely arranged and reduced in number, as shown by the blue arrowheads; the lymphocytes in the red medulla were fewer in number and sparsely arranged, as shown by the black arrowheads (Figure 2D, Appendix A). This study indicates that, compared to the NC group, both Quantity of white pulp and Average area of white pulp were significantly reduced in the H3, H5, and H7 groups (Figure 2E,F).

#### 2.1.3. Effects of Different Hypobaric Hypoxia Times on Mouse Spleen Immune Cells

To further verify the effect of the plateau immune environment on immune cells, immune cells in the spleen were analyzed by flow cytometry (Appendix A). With the increase in hypoxia time, the number of CD45^+^ cells in the spleen showed a tendency to decrease and then increase, with a significant decrease after day 3 and the lowest in the H5 group (Figure 3A,D); the number of CD3^+^ T cells in the spleen showed a significant decrease after day 5 (Figure 3B,E); and there was no significant change in the number of CD19^+^ B cells in the spleen (Figure 3B,F). It was found that acute plateau hypobaric hypoxic stimulation resulted in an elevated or unchanged B-cell population [21], whereas chronic plateau hypobaric hypoxic stimulation resulted in a decreased B-lymphocyte population [22]. Representative images of CD4 and CD8 cells in the spleen cells measured by flow cytometry (Figure 3C). CD3^+^CD4^+^ T-cells were significantly reduced in the spleen (Figure 3G); CD3^+^CD8^+^ T cells in the spleen were elevated and then normalized, significant only in H1 and H3 (Figure 3H). The CD4^+^/CD8^+^ cell ratios in the spleen were all significantly decreased and had a tendency to recover with time (Figure 3I).

#### 2.1.4. Effects of Different Hypobaric Hypoxia Times on the Transcriptome and Bioinformatics of Mouse Spleen

To investigate the effects of different durations of hypobaric hypoxia on gene expression, spleen tissues from normal control and hypobaric hypoxia control mice were isolated, RNA was extracted, and high-throughput sequencing was performed on the DNBSEQ-T7 platform. Between 44 and63 million raw sequence reads were obtained for each sample. DEG-based clustering analysis of the samples showed that the response to exposure to hypobaric hypoxia varied over time (Figure 4A). UMAP plots showed that samples were clustered within groups and dispersed between groups (Figure 4B), suggesting that changes in transcriptional expression of genes occurred consistently in samples within groups and showed differentiation between groups. A total of 1341 DEGs were found in the spleen tissues of group H1, 2657 DEGs were found in group H3, 3622 DEGs were found in group H5, and 2780 DEGs were found in the H7 group (Figure 4C).

According to the differential gene Venn diagram (Figure 4D), there were 531 DEGs overlapping in the four groups; according to the down-regulated differential gene Venn diagram (Figure 4E), there were 271 DEGs overlapping in the four groups; and according to the up-regulated differential gene Venn diagram (Figure 4F), there were 260 DEGs overlapping in the four groups. In order to elucidate the potential functions of the 531 key genes, KEGG pathway enrichment analysis was performed, and the relevant genes were mainly enriched in the following pathways: cytokine-cytokine receptor interaction, mineral absorption, primary immunodeficiency, etc. (Figure 4G, Appendix A). Down-regulated genes were mainly enriched in cytokine-cytokine receptor interaction, primary immunodeficiency, natural killer cell-mediated cytotoxicity, NF-kappa B signaling pathway, Th1 and Th2 cell differentiation, etc. (Figure 4H, Appendix A). Up-regulated genes were mainly enriched in mineral absorption, DNA replication, sphingolipid metabolism, etc. (Figure 4I, Appendix A). Through the enrichment analysis, it was found that the immunoinflammation-related signaling pathways were mainly down-regulated.

### 2.2. Regulatory Effects of AS-IV on the Hypobaric Hypoxia Immune Injury Model

#### 2.2.1. Network Pharmacology of AS-IV on Immune Impairment

A total of 514 potential targets of AS-IV were collected through a network pharmacology approach. A total of 364 immune-related targets were obtained through the GeneCards database. Finally, 25 targets were considered to be common targets of AS-IV and immunity (Appendix A), which were screened for further KEGG pathway enrichment analysis (Appendix A). The results of KEGG pathway enrichment analysis showed that the potential targets were associated with immunologically relevant Th17 cell differentiation, Toll-like receptor signaling pathway, Th1 and Th2 cell differentiation, and Chemokine signaling pathway (Appendix A).

#### 2.2.2. Effects of AS-IV on Splenic Indices and Pathology in a Plateau Hypoxic Immune Injury Model

We found that the signaling pathways of immune injury induced by hypobaric hypoxic stimulation and the regulation of immunity by AS-IV were related to the Th1 and Th2 cell differentiation pathway through pre-transcriptomics and network pharmacology, so we focused on the Th1 and Th2 cell differentiation for in-depth study to explore the lymphocyte regulatory effects of AS-IV on mice with hypobaric hypoxic immune injury model. regulation, and designed animal experiments in mice (Figure 5A). Compared with the control group, the body weight of mice in the model group was significantly reduced, and the administration of the drug significantly restored the body weight of mice (Figure 5B,C). Compared with the control group, the spleen coefficients of mice in the model group were all significantly elevated, and compared with the model group, the spleen coefficients of mice were significantly reduced after AS-IV treatment (Figure 5E). In the NC group, the spleen tissue structure was basically normal. In the model group, the splenic tissues of mice showed heavy structural abnormalities, with scattered splenic nodule structures in the field of view, blurred boundaries between red and white medulla, reduced white medulla area, decreased number of small lymphocytes as shown by the yellow arrows, and a small amount of inflammatory cell infiltration was seen in the tissues as shown by the black arrows. Splenic tissues of mice in the AS-IV treatment group showed mild structural abnormalities, with a decrease in the scattered structure of splenic knots in the field of view, a clear boundary between red and white medulla, an increase in the number of small lymphocytes compared with the model group as shown by the yellow arrowheads, and a decrease in the infiltration of inflammatory cells in the tissues as shown by the black arrowheads (Figure 5D, Appendix A). Meanwhile, compared with the control group, the number and area of white medulla in the model group were significantly reduced, and the number of white medulla could be significantly restored after AS-IV treatment (Figure 5F,G). This suggests that the pathological changes of the spleen induced by high-altitude hypobaric hypoxia can be improved by pretreatment with AS-IV.

#### 2.2.3. Effects of AS-IV on Physiological Indexes in Plateau Hypoxic Immune Injury Model

Compared to the control group, the model group had elevated WBC content (Figure 5H), significantly lower LYMPH% (Figure 5I), and significantly higher NEUT% (Figure 5J), and administration of the drug reversed these differences. In addition, RBC, BASO%, and HGB levels were significantly higher and MONO%, EO%, and PLT were significantly lower in the model group compared to the control group, and AS-IV intervention reversed these differences with a reduction in treatment (Appendix A).

#### 2.2.4. Effects of AS-IV on Peripheral Blood and Spleen Lymphocytes in a Plateau Hypoxic Immune Injury Model

To further determine the regulatory effects of AS-IV on lymphocytes in plateau hypoxic immune injury, CD3^+^ T, CD3^+^CD4^+^ T, and CD3^+^CD8^+^ T immune cells and CD4+/CD8+ cell ratios were examined in peripheral blood (Figure 6A–E) and spleen (Figure 6F–J) of mice. The results showed that astragalus methylglycoside had a significant modulating effect on CD3^+^ T, CD3^+^CD4^+^ T, and CD4^+^/CD8^+^ cells in peripheral blood and spleen in plateau hypoxic immune injury model mice.

In order to further verify the effect of AS-IV on the Th1/Th2 cell pathway, the cytokines of Th1 (IFN-γ) and Th2 (IL-4) were detected in the spleen of mice (Figure 7A,C). CD4 is a marker of T helper cells, used to distinguish T cell subsets. IFN-γ and IL-4 are the specific cytokines of Th1 and Th2 cells, respectively, which clearly differentiate their functional characteristics. By detecting CD4^+^ IFN-γ^+^ and CD4^+^ IL-4^+^ cells using flow cytometry, Th1 and Th2 cell populations can be accurately identified. It was found that Th1 (CD4^+^ IFN-γ^+^) cells were significantly decreased in the spleen in plateau hypoxia (Figure 7B), and significantly increased after drug administration; plateau hypoxia did not have a significant effect on Th2 (CD4^+^ IL-4^+^) cells (Figure 7D), which might be related to the low expression of IL-4. It indicated that AS-IV could regulate the imbalance of Th1 and Th2 cells caused by plateau hypoxia.

#### 2.2.5. Effects of AS-IV on Reactive Oxygen Species and Apoptotic Cells of Spleen Cells in a Plateau Hypoxic Immune Injury Model

To further determine the mechanism of plateau immune injury, reactive oxygen species and apoptosis were examined in mouse spleen cells. Compared with the control group, the reactive oxygen species was significantly elevated in the spleen cells in the model group and significantly decreased after drug administration (Figure 7E,F), indicating that AS-IV significantly alleviated the increase in the reactive oxygen species in the spleen caused by plateau hypoxia. Compared with the NC group, the degree of apoptosis of splenocytes in the model group was significantly increased and significantly decreased after drug administration (Figure 7G,H), indicating that AS-IV could significantly alleviate the increase in the apoptosis of spleen caused by plateau hypoxia.

#### 2.2.6. Effect of AS-IV on Peripheral Blood Serum Inflammatory Factors in a Plateau Hypoxic Immune Injury Model

In addition, we analyzed the expression of inflammatory factors IL-2, IL-4, IL-6 and IL-10 in serum. Compared with the NC group, IL-2 and IL-6 were significantly elevated and IL-4 and IL-10 were significantly decreased in the serum of mice in the model group; compared with the model group, IL-2 and IL-6 were significantly decreased, and IL-4 and IL-10 were significantly elevated after AS-IV administration (Figure 8A–D). The results indicated that AS-IV had a modulating effect on the elevation of pro-inflammatory factors IL-2 and IL-6, and the reduction of anti-inflammatory factors IL-4 and IL-10 in the serum of plateau immunoinjured mice, which proved that pretreatment with AS-IV might modulate plateau immunoinjury by inhibiting inflammatory responses.

#### 2.2.7. Effects of AS-IV on Spleen mRNA in a Plateau Hypoxic Immune Injury Model

To further investigate the regulatory role of AS-IV on the Th1/Th2 signaling pathway in the spleen under high-altitude hypoxia, the mRNA expression levels of the following genes in the Th1/Th2 signaling pathway were measured: *Makp11*, *Bcl6*, *Lck*, *Ifng*, *Cd3b*, *Cd3e*, *Cd3g*, *Cd4*, *Runx1*, *Runx3*, *Il12rb2*, *H2-Ab1*, *H2-Ob*, *Cd247*, *Tbx21, Gata3, H2-Oa*, *Il12a*, *Il4ra*, and *H2-Aa*. Compared with the NC group, the mRNA expression of *Makp11*, *Bcl6*, *Lck*, *Cd3b*, *Cd3e*, *Cd3g*, *Cd4*, *Runx1*, *Runx3*, *Il12rb2*, and *H2-Ab1* was significantly decreased in the model group, whereas AS-IV significantly increased their expression (Figure 8E–L, Appendix A). As for the mRNA expression of *Tbx21, Gata3, H2-Ob*, *Cd247*, *H2-Oa*, *Il12a*, *Il4ra*, and *H2-Aa*, plateau hypoxia significantly reduced their expression, but AS-IV did not have a significant regulatory effect (Appendix A). It was demonstrated that AS-IV could regulate plateau immune injury through the Th1/Th2 signaling pathway.

## 3. Discussion

High-altitude hypobaric hypoxia environments may lead to damage to the immune system, which is usually associated with immune cell dysfunction and increased inflammatory responses [20]. However, the specific effects of hypobaric hypoxia on the immune system and its pathogenesis are not yet fully understood. The aim of this study was to explore the optimal animal modeling conditions for inducing hypobaric hypoxia immune injury in mice by comparing the changes in immune status under simulated hypobaric hypoxia conditions at an altitude of 7000 m. Specifically, this study was designed to observe the proportion of immune cells in the peripheral blood of mice as well as the changes in splenic pathology at different hypoxia time points (1 day, 3 days, 5 days, and 7 days). At the same time, the key signaling pathways affecting hypobaric hypoxia immune injury were explored through transcriptomic analysis, and then experimental intervention was carried out by AS-IV to study its preventive and therapeutic effects on hypobaric hypoxia immune injury.

Studies have shown that hypoxia can lead to inflammatory infiltration and inflammatory response of various types of immune cells in mice [23]. The size and physiological function of the spleen can affect the number and function of T lymphocytes to a certain extent [24]. In this study, it was found that spleen enlargement was most significant on the fifth day under hypobaric and hypoxic conditions at a simulated altitude of 7000 m. Blood routine and spleen HE results showed that the number of lymphocytes decreased, while the number of neutrophils increased, and both reached the maximum value on day 5. T lymphocyte subsets are a key indicator of cellular immune function and are the main immune effector cells in the body [25]. CD4^+^ T and CD8^+^ T cells, play a central role in maintaining the immune balance in the body and determine the stability of the immune environment in the body. Under normal circumstances, the CD4^+^/CD8^+^ cell ratio remains relatively stable [26]. Studies have shown that after 1 day and 3 days of acute hypoxia, the number of CD3^+^ T and CD4^+^ T cells in peripheral blood gradually decreased with the increase in time [19]. For people living in plateau areas for a long time, the number of CD4^+^ T cells and the ratio of CD4^+^/CD8^+^ cells decreased, while the number of CD8^+^ T cells increased [22]. In this study, the number of CD45^+^ T, CD3^+^ T, and CD3^+^CD4^+^ T cells and the ratio of CD4^+^/CD8^+^ cells in the spleen of mice were decreased. These results indicate that high-altitude hypoxia can cause immune damage and inhibit the function of T lymphocytes in mice. Meanwhile, studies have shown that in the acute altitude hypoxia environment, the expression of immune-related signaling pathways in human serum proteomics and metabolomics is reduced, especially the T-lymphocyte related signaling pathway, which indicates that altitude hypobaric hypoxia will lead to the immune dysfunction of the body [27]. In this study, RNA sequencing analysis showed that the number of DEGs in the H5 group reached a peak and then declined. This phenomenon may reflect the process of mice gradually adapting to the hypobaric hypoxia environment. Further, the down-regulated KEGG pathway enrichment analysis showed that Th1 and Th2 cell differentiation and the chemokine signaling pathway were the most significant signaling pathways for immune injury at high altitudes, indicating immune dysfunction after exposure to hypobaric hypoxia at high altitudes [27]. With the deepening of the study of cellular immune function, it has been found that CD4^+^ T cells can be further divided into multiple subtypes, and activated Th cells are divided into Th1, Th2, and Th17 subgroups according to the different cytokines they secrete [25]. If the immune regulation of the body’s Th1/Th2 cells is out of balance, it will exacerbate the inflammatory response. Studies have shown that AS-IV can target lymphocytes by regulating TLR4/NF-kB, PI3K-AKT and other pathways to regulate immune function [28,29]. In a mouse model of kidney injury in IgA nephropathy, AS-IV can regulate the Th1/Th2 cell ratio by up-regulating the level of IFN-γ [30]. This study showed that AS-IV can significantly reduce splenomegaly, increase the lymphocyte content in pathological tissue of the spleen, and decrease the neutrophil content. At the same time, it significantly increased the number of CD4^+^ T cells and the CD4^+^/CD8^+^ cell ratio in the spleen and peripheral blood. Further studies found that AS-IV significantly increased Th1 (CD4^+^ IFN-γ^+^) cell content, indicating that AS-IV can regulate Th1/Th2 cell balance by increasing Th1 cell content, thereby increasing the number of CD4^+^ T cells.

Excessive secretion of cytokines and inflammatory mediators can aggravate oxidative stress damage in the body. If the disease progresses further, an imbalance in the proportion of Th cells may lead to immunosuppressation [31]. Previous studies have shown that acute altitude hypoxia environmental triggers the up-regulation of inflammatory signaling pathways, resulting in increased expression of inflammatory genes associated with immune sensitization [27]. Studies have shown that reduced CD4^+^/CD8^+^ ratios, resulting from decreased CD4^+^ T cells and increased CD8^+^ T cells, may activate inflammatory responses in people at high elevations [22]. In addition, acute hypoxia leads to increased serum levels of pro-inflammatory factors (such as IL-1β, IL-6, TNF-α) and decreased levels of anti-inflammatory factors (such as IL-4, IL-10) [32]. This study shows that the level of pro-inflammatory factors (such as IL-2 and IL-6) increases and the level of anti-inflammatory factors (such as IL-4 and IL-10) decreases under the hypoxia environment at high altitudes, which indicates that hypoxia at high altitudes will aggravate the inflammatory response of the body. Studies have shown that AS-IV can significantly reduce the levels of ROS, IL-6, and TNF-α in Beas-2B cells and vascular endothelial cells induced by intermittent hypoxia, thereby reducing inflammation and oxidative stress levels [33]. In addition, AS-IV can improve the oxidative stress and apoptosis of hippocampal neurons caused by long-term hypoxia [34], and significantly reduce the apoptosis rate and inflammatory damage in lung epithelial cells during hypoxia injury [35]. Through the MAPK pathway, AS-IV can reduce Annexin V levels and increase Bcl6 levels, thereby inhibiting apoptosis [28]. This study found that AS-IV can significantly reduce the levels of pro-inflammatory factors such as IL-2 and IL-6, and increase the expression of anti-inflammatory factors such as IL-4 and IL-10 caused by high altitude hypoxia. This bidirectional regulation effect is helpful in restoring immune homeostasis and reducing inflammatory damage under high altitude hypoxia. At the same time, a significant increase in ROS levels in the spleen was observed under high-altitude hypoxia, which may further damage cell function. However, AS-IV attenuates cell damage caused by oxidative stress by reducing ROS levels, thereby indirectly inhibiting the exacerbation of inflammatory responses. It was further found that AS-IV could significantly alleviate the increase in apoptosis induced by high altitude hypoxia. AS-IV significantly reduced apoptosis induced by high altitude hypoxia by up-regulating Bcl6 expression and inhibiting Annexin-V level, and further maintained the function and quantity of immune cells. In addition, AS-IV significantly up-regulated the mRNA expression of *Bcl6* and *Mapk11*, as well as Th1/Th2 cell differentiation pathway related genes (such as *Cd3g*, *Cd3e*, *Cd3*, *Cd4*, *Ifng*). These results indicate that AS-IV can inhibit cell apoptosis and inflammatory response by activating MAPK pathway (such as up-regulating *Mapk11* expression), and further protect cells from hypoxia at high altitude. Literature studies have shown that detecting the mRNA levels of Th1/Th2 signaling pathway-related genes in the spleen can further confirm the close association between the imbalance of Th1/Th2 cell equilibrium and splenic immune dysfunction [36,37]. This study demonstrate that AS-IV treatment can ameliorate abnormalities in inflammatory factors within the Th1/Th2 cell pathway in the spleen of mice induced by hypobaric hypoxia, but its regulatory effects on transcription factors are limited.

In conclusion, AS-IV pretreatment plays an important protective role in immune injury at high altitudes by inhibiting inflammatory response, alleviating oxidative stress and regulating apoptosis. Its multi-target and multi-pathway mechanism of action provides a broad prospect for its application in high-altitude medicine and the treatment of inflammation-related diseases. Future research should further explore its clinical translation potential to provide new solutions for health protection and inflammation treatment at high altitudes.

## 4. Materials and Methods

### 4.1. Mice Model

Male BALB/c mice, weighing 22 ± 2 g, were purchased from Beijing Vital River Laboratory Animal Technology Co., Ltd, Beijing, China. Animal Production License number: SCXK(Beijing)2021-0006. The mice were housed in micro-isolation cages under a 12 h light-dark cycle in standard laboratory conditions with a temperature of 25 ± 1 °C and humidity of 60% ± 5%. They had unrestricted access to food and water. All animal experiments were approved by the Institutional Animal Care and Use Committee (IACUC) of the Academy of Military Medical Sciences, Beijing, China, with ethics approval number IACUC-DWZX-2023-564.

Fifty male BALB/c mice were randomly divided into five groups (n = 10 per group): Plain Normoxia Control Group (NC), High Altitude Hypoxia 1 Day Group (H1), High Altitude Hypoxia 3 Days Group (H3), High Altitude Hypoxia 5 Days Group (H5), High Altitude Hypoxia 7 Days Group (H7). The NC group was housed under similar conditions of light, temperature, and humidity but in a separate room maintained at normal atmospheric pressure and normoxic conditions. Mice in the hypobaric hypoxia groups were placed in a hypobaric hypoxia chamber (Guizhou Fenglei Aviation Ordnance Co. Ltd., Guizhou, China, CAT: DWCF50-IIA) and exposed to a simulated altitude of 7000 m (308 mmHg, equivalent to Pio2 8.0 kPa) at a rate of 10 m/s. Samples were collected from the mice after 1, 3, 5, and 7 days of exposure to hypobaric hypoxia.

Eighteen male BALB/c mice were randomly divided into three groups: normoxia control group (NC group), model group (MOD group), and astragaloside IV group (AS-IV group), with six mice in each group. AS-IV group: mice were gavaged with 80 mg/kg AS-IV daily for 12 days. NC group and MOD group were given an equal volume of sodium carboxymethyl cellulose by gavage every day for 12 consecutive days. Each group was given prophylaxis on days 0–7. From day 8, the mice in the MOD and AS-IV groups were placed in a hypobaric hypoxia chamber and exposed to a simulated hypoxia environment at an altitude of 7000 m (308 mmHg, equivalent to Pio2 8.0 kPa) at a rate of 10 m/s. Treatment was performed by intragastric administration after daily destocking. Light, temperature, and humidity conditions were similar in the NC group, but normal atmospheric pressure and normoxia conditions were maintained in a separate room. Sampling and testing were performed on day 12.

### 4.2. Chemical Reagents and Instruments

Fluorescence quantitative PCR instrument (Bio-Rad Laboratories, Inc., Hercules, CA, USA, CAT:CFX Opus 96); flow cytometer (Becton, Dickinson and Company, Franklin Lakes, NJ, USA, CAT: FACSAria™ III); blood cell analyzer (Sysmex Corporation, Kobe, Japan, CAT: XN-1000V); astragaloside IV (Shanghai Yuanye Biotechnology Co., Ltd., Shanghai, China, CAT: B20564); flow antibody 7-ADD viability staining solution (BioLegend, Inc., San Diego, CA, USA, CAT: 420404), APC-cy7 anti-mouse CD45 (Biolgend, CAT: 103116), FITC anti-mouse CD3 (Biolegend, CAT: 100204), PE anti-mouse CD4 (Biolegend, CAT: 100408), PE-cy7 anti-mouse CD8 (Biolegend, CAT: 100722), APC anti-mouse CD19 (Biolegend, CAT: 115512), Alexa Fluor^®^ 700 anti-mouse CD3 (Biolegend, CAT: 100215), FITC anti-mouse CD4 (Biolegend, CAT: 100509), PerCP/Cyanine5.5 anti-mouse CD8a (Biolegend, CAT: 100733), PE/Cyanine7 anti-mouse IFN-γ (Biolegend, CAT: 505825), PE anti-mouse IL-4 (Biolegend, CAT: 504103), zombie aqua™ fixable viability kit (Biolegend, CAT: 423101), trustain fcx™ (anti-mouse CD16/32) (Biolegend, CAT: 101319), cell staining buffer (Biolegend, CAT: 420201), cell activation cocktail (with Brefeldin A) (Biolegend, CAT: 423303), true-nuclear™ transcription factor buffer set (Biolegend, CAT: 424401); erythrocyte lysate (Beijing Solarbio Science & Technology Co., Ltd., Beijing, China, CAT: R1010); interleukin-2 (IL-2) (Meimian Industrial Co., Ltd., Jiangsu, China, CAT: MM-0701M1), interleukin-4 (IL-4) (Meimian Industrial Co., Ltd., CAT: MM-0165M1), interleukin-6 (IL-6) (Meimian Industrial Co., Ltd., CAT: MM-0163M1, interleukin-10 (IL-10) (Meimian Industrial Co., Ltd., CAT: MM-0176M1); reverse transcription kit (YEASEN, Shanghai, China, CAT: 11123ES60), qPCR mix (YEASEN, Shanghai, China, CAT: 11201ES08,); trizol (Sigma-Aldrich, St. Louis, MO, USA, CAT: 102663540,); cellular reactive oxygen species kit (Beyotime Biotechnology, Shanghai, China, CAT: S0033M), apoptosis kit (Becton, Dickinson and Company, Franklin Lakes, NJ, USA, CAT: 559763), and primers (Sangon Biotech, Shanghai, China).

### 4.3. Sample Collection

Throughout the experiment, the body weight of the mice was measured daily. After successful modeling, whole blood samples were collected from the orbital venous plexus before euthanasia. A portion of the blood was collected into ethylenediaminetetraacetic acid (EDTA) anticoagulant tubes for complete blood count or flow cytometry analysis of immune cells, while another portion was collected into non-anticoagulant tubes for ELISA analysis. The non-anticoagulant blood samples were kept at 4 °C for 30 min and then centrifuged at 1500× *g* for 15 min. The supernatant (serum) was collected and stored at −20 °C until analysis. Spleen samples were promptly weighed and collected for analysis of morphological changes, oxidative indicators, cell apoptosis, and relative mRNA expression.

### 4.4. Blood Routine Tests

Complete blood counts were performed using anticoagulated whole blood with a hematology analyzer.

### 4.5. Spleen Index

Mice were euthanized by cervical dislocation. The abdominal area was disinfected with alcohol, and the abdominal cavity was opened using ophthalmic scissors to retrieve the spleen. The organ surface blood was blotted dry with filter paper, and the spleen was then weighed. The spleen index was calculated as follows: Spleen Index = Spleen Weight/Body Weight (g/g).

### 4.6. Flow Cytometry

Single cell suspensions for flow cytometry were prepared from spleen and peripheral blood, and the cell suspension was adjusted to 1 × 10^6^ cells /mL. Cells were collected by centrifugation and resuspended in 100 μL PBS (Gibco, Grand Island, NY, USA, CAT: 10010023). Anti-mouse CD45 APC-Cy7, CD3 FITC, CD4 PE, CD8 PE-Cy7, and CD19 APC antibodies were added. The plates were incubated for 15 min at room temperature in the dark. After incubation, cells were washed twice with PBS and resuspended in 400 μL of PBS. The stained cell suspension was transferred to a flow cytometer tube for analysis to determine the percentage of T and B lymphocytes.

The spleen cell suspension was collected at 1 × 10^6^ cells /mL, stimulated and incubated with a mixture of PMA/Ionomycin and BFA/Monensin at 37 °C with 5% CO_2_ for 5 h. Surface marker staining was performed by incubating the cells with FITC-anti-CD4 antibody in the dark at 4 °C for 15 min. The cells were then fixed and permeabilized using a Zombie Aqua™ Fixable Viability Kit (BioLegend, Inc., San Diego, CA, USA, CAT: 423101). Intracellular staining was conducted by incubating the cells with PE-Cy7 anti-IFN-γ antibody and PE-anti-IL-4 antibody for 0.5 h in the dark at room temperature. Finally, the cells were analyzed by flow cytometry. IFN-γ and IL-4 were used to determine Th1 and Th2 cells.

### 4.7. ROS Detection

Reactive oxygen species (ROS) were detected using the FITC ROS Detection Kit. Spleen cells were resuspended in serum-free RPMI 1640 medium (Gibco, Grand Island, NY, USA, CAT: 11875093) and incubated in a 37 °C incubator for 20 min. The cells were then washed twice with PBS, and the supernatant was discarded. Finally, the cells were resuspended in 300 μL of PBS.

### 4.8. Pharmingen PE Annexin V Apoptosis Detection

Apoptosis was detected using the Pharmingen PE Annexin V Apoptosis Detection Kit (Becton, Dickinson and Company, Franklin Lakes, NJ, USA, CAT: 559763). Spleen single-cell suspensions were washed with PBS and centrifuged at 4 °C for 5 min at 1500 r/min. The supernatant was discarded, and the cells were resuspended in 100 μL of 1× FITC Binding Buffer. Cells were then stained with 5 μL of FITC Annexin V and 5 μL of 7-AAD in the dark for 15 min. After staining, 400 μL of 1× FITC Binding Buffer was added, and the lymphocyte apoptosis rate was analyzed by flow cytometry.

### 4.9. Histopathological Analysis

The collected spleen tissues were fixed in 4% paraformaldehyde, dehydrated, and embedded in paraffin. Paraffin sections approximately 5 µm thick were prepared and subjected to dewaxing and rehydration steps prior to staining. The sections were placed in a hematoxylin staining solution for 10 to 15 min, followed by rinsing with distilled water. The sections were then stained in an eosin staining solution for 15 s and washed with distilled water. Dehydration was performed using a graded ethanol series (75%, 85%, 95%, and 100%) for 2 min each, followed by clearing in xylene. The sections were then sealed with a neutral mounting medium and removed from the water at room temperature. Finally, the sections were sealed with a neutral mounting medium and allowed to dry at room temperature. The number and area of white pulp were measured with image analysis software, such as ImageJ 1.51j8, and statistical analysis was performed.

### 4.10. RNA Preparation and Sequencing

Following exposure to a simulated hypobaric and hypoxic environment equivalent to an altitude of 7000 m, mRNA profiles of mouse spleen tissues were generated using deep sequencing with three biological replicates. The data were preprocessed by sequentially removing adapter sequences and low-quality reads, trimming low-quality bases from the 3′ or 5′ ends, analyzing raw and clean sequencing reads, and performing guanine and cytosine content statistics. Differential expression analysis was conducted using correlation tests, principal component analysis (PCA), and clustering analysis to identify differentially expressed genes (DEGs). Genes with a fold change (FC) greater than 2 and a *p*-value ≤ 0.05 were considered significantly differentially expressed. These DEGs were then mapped to pathways in spleen tissue using the Kyoto Encyclopedia of Genes and Genomes (KEGG) database.

### 4.11. Enzyme-Linked Immunosorbent Assay (ELISA)

Serum cytokine levels of IL-2, IL-4, IL-6, and IL-10 were measured using enzyme-linked immunosorbent assay (ELISA) kits (Meimian Industrial Co., Ltd., Jiangsu, China) according to the manufacturer’s instructions. Serum supernatants from the mice were used for these assays.

### 4.12. Quantitative Real-Time Quantitative Polymerase Chain Reaction (qRT-PCR)

Total RNA was extracted and purified from mouse spleen tissues using Trizol (Sigma-Aldrich, St. Louis, MO, USA) reagent. The total RNA was then reverse transcribed into cDNA using a one-step reverse transcription PCR kit according to the manufacturer’s instructions. Quantitative real-time PCR (qPCR) was performed using the CFX 96 RT PCR Detection System (Bio-Rad Laboratories, Inc., Hercules, CA, USA) with a specific qPCR kit and the cDNA as the template, following the manufacturer’s protocol. Primer sequences are listed in Table 1 and Appendix A.

### 4.13. Network Pharmacology Analysis

To obtain comprehensive information on AS-IV and its structural details, data were collected from multiple databases: Traditional Chinese Medicine Information Platform (TCMIP): (http://www.tcmip.cn/ETCM/index.php/Home/, accessed on 15 October 2024), Bioinformatics Analysis Tool for Molecular mechanisms of TCM (BATMAN-TCM) (http://bionet.ncpsb.org.cn/batman-tcm/, accessed on 15 October 2024), PubChem: (https://pubchem.ncbi.nlm.nih.gov/, accessed on 15 October 2024). Potential targets of AS-IV were predicted using: Swiss Target Prediction: (http://swisstargetprediction.ch/, accessed on 15 October 2024), PubChem: (https://pubchem.ncbi.nlm.nih.gov/, accessed on 15 October 2024), Immune-related targets were identified in the GeneCards database with a relevance score ≥ median: (https://www.genecards.org/, accessed on 15 October 2024). All targets were standardized and validated using the UniProt database, selecting Homo sapiens as the target organism: (https://www.uniprot.org/, accessed on 15 October 2024). Venny 2.1 was used to analyze and visualize the overlap of targets obtained from the above databases: (http://bioinfogp.cnb.csic.es/tools/venny/index.html, accessed on 15 October 2024). Protein-protein interaction (PPI) networks were constructed using the STRING database: (https://cn.string-db.org/, accessed on 15 October 2024) and visualized with Cytoscape 3.9.1. Core targets were screened based on network topology analysis. Finally, the Metascape database was utilized for gene ontology (GO) enrichment analysis and Kyoto Encyclopedia of Genes and Genomes (KEGG) pathway enrichment analysis of the core targets to investigate the underlying biological processes and signaling pathways: (https://metascape.org/gp/index.html, accessed on 15 October 2024).

### 4.14. Statistical Analysis

All data are expressed as mean ± SD. Statistical analyses were performed using GraphPad Prism version 9 (GraphPad Software, La Jolla, CA, USA). For comparisons among multiple groups, one-way ANOVA was used, while for comparisons between two groups, a *t*-test was employed. Statistical significance in all figures is indicated as follows: * *p* < 0.05; ** *p* < 0.01; *** *p* < 0.001; **** *p* < 0.0001. 

## 5. Conclusions

High altitude hypobaric hypoxia can lead to splenomegaly, reduce lymphocyte counts and the Th1/Th2 cell differentiation pathway, thereby inducing immune injury. This study revealed that AS-IV alleviated high-altitude hypoxia-induced immune injury by regulating the Th1/Th2 cell differentiation pathway and related genes, attenuating splenic pathological injury, and inhibiting oxidative stress and inflammatory responses. This study provides a new idea for drug research on high-altitude hypobaric hypoxia-associated immune injury.

## Figures and Tables

**Figure 1 ijms-26-02584-f001:**
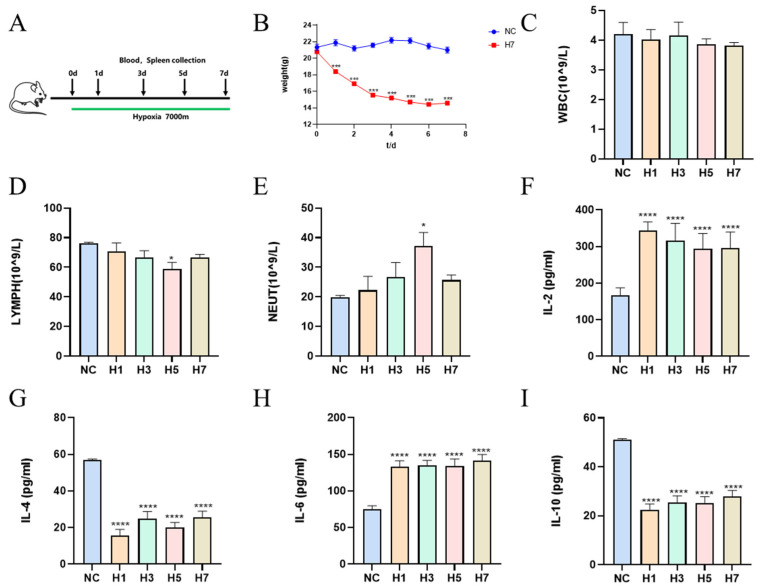
Effects of different hypobaric hypoxia times on physiological and biochemical indexes. (**A**) High altitude immune damage model modeling scheme. (**B**) Changes in mouse body weight (*n* = 10). (**C**–**E**) Changes of main indexes of blood routine tests (*n* = 4–6). (**F**–**I**) Changes of main indexes of cytokine (*n* = 8). * *p* < 0.05, *** *p* < 0.001, **** *p* < 0.0001 vs. NC.

**Figure 2 ijms-26-02584-f002:**
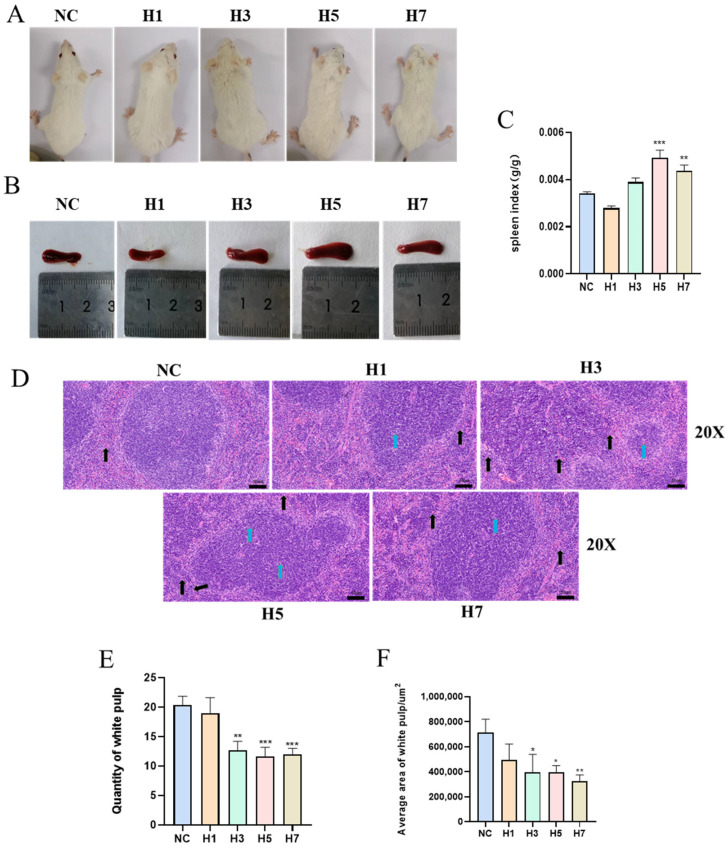
Effect of different hypobaric hypoxia times on histopathology and viscera-containing coefficient of the spleen. (**A**) Image of the mouse. (**B**) Image of the spleen. (**C**) Spleen index (*n* = 8–10). (**D**) Histopathological changes of spleen tissues by H&E staining in each group, magnification 20× (*n* = 3). Following exposure to hypoxic conditions, splenic tissue exhibited abnormalities, such as a decrease in lymphocyte numbers (black arrow). In addition, an increased number of neutrophils in the spleen tissue showed inflammatory infiltration (blue arrow) (The high-resolution images are available in Appendix A). (**E**) Quantity of white pulp in the spleen (*n* = 3). (**F**) Average area of white pulp in the mouse (*n* = 3). * *p* < 0.05, ** *p* < 0.01, *** *p* < 0.001 vs. NC.

**Figure 3 ijms-26-02584-f003:**
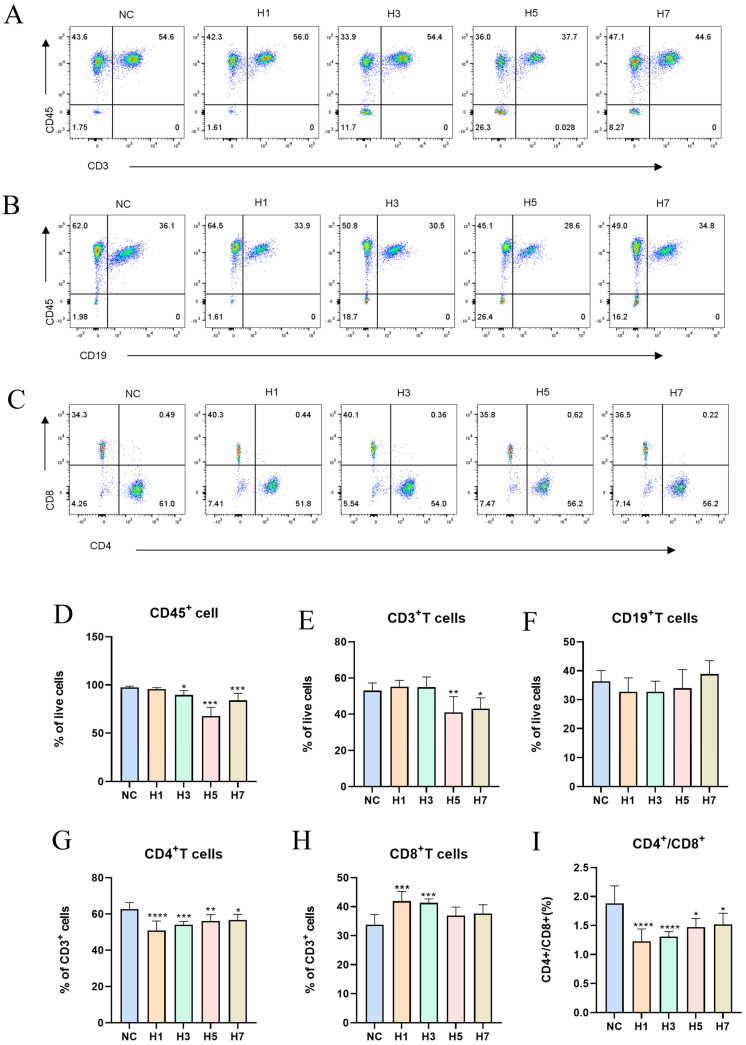
Effects of different hypobaric hypoxia times on immune cells in the spleen. (**A**) Representative images of CD45 and CD3 cells in the spleen cells measured by flow cytometry. (**B**) Representative images of CD45 and CD19 cells in the spleen cells measured by flow cytometry. (**C**) Representative images of CD8 and CD4 cells in the spleen cells measured by flow cytometry. (**D**) CD45^+^ immune cell (*n* = 4–6). (**E**) CD3^+^ T-lymphocyte (*n* = 4–6). (**F**) CD19^+^ B-lymphocyte (*n* = 4–6). (**G**) CD4^+^ T-lymphocyte (*n* = 4–6). (**H**) CD8^+^ T-lymphocyte (*n* = 4–6). (**I**) CD4^+^/CD8^+^ ratio (*n* = 4–6). * *p* < 0.05, ** *p* < 0.01, *** *p* < 0.001, **** *p* < 0.0001 vs. NC.

**Figure 4 ijms-26-02584-f004:**
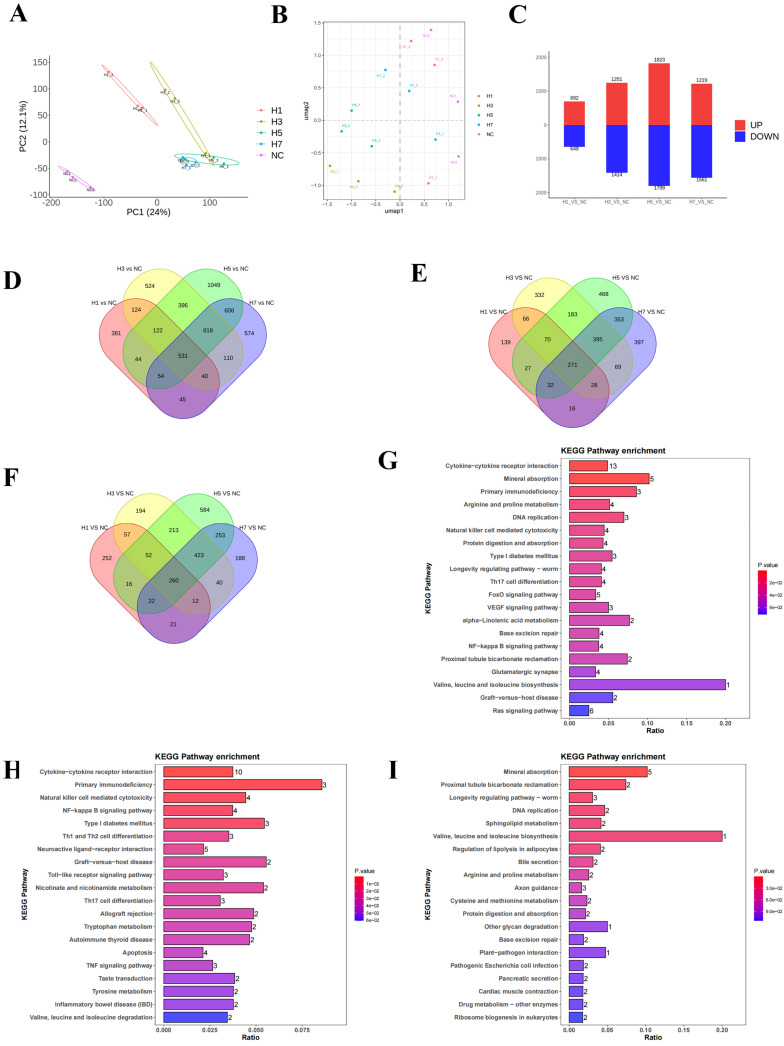
Effects of different hypobaric hypoxia durations on spleen transcriptome and bioinformatics. (**A**) Clustering of transcriptomes using PCA (*n* = 3). (**B**) Clustering of transcriptomes using UMAP (*n* = 3). (**C**) Quantitative analysis of genes significantly up- and down-regulated in each group (*n* = 3). (**D**) Venn diagram comparing the significant DEGs in each group. (**E**) Venn diagram comparing the significant DEGs of down-regulated genes in each group. (**F**) Venn diagram comparing the significant DEGs of up-regulated genes in each group. (**G**) KEGG enrichment analysis with the 20 most enriched KEGG terms shown in all DEGs (The high-resolution images are available in Appendix A). (**H**) KEGG enrichment analysis with the 20 down-regulated most enriched KEGG terms shown in all DEGs (The high-resolution images are available in Appendix A). (**I**) KEGG enrichment analysis with the 20 up-regulated most enriched KEGG terms shown in all DEGs (The high-resolution images are available in Appendix A).

**Figure 5 ijms-26-02584-f005:**
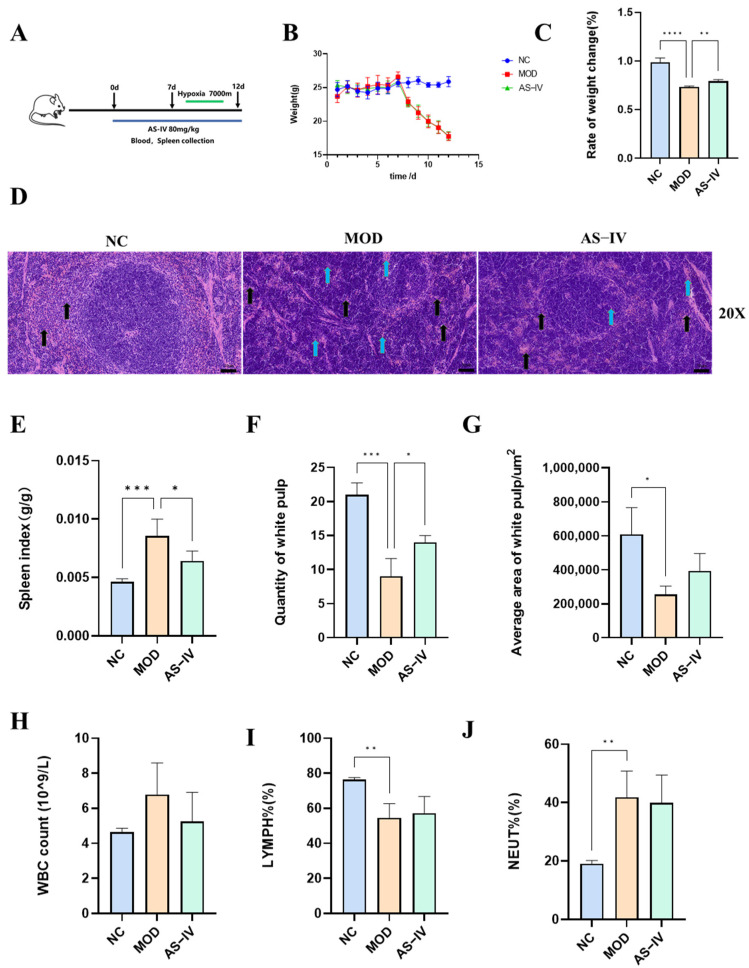
Effects of AS-IV on body weight, spleen and blood routine in mice with high altitude immune injury. (**A**) High altitude immune injury model and AS-IV administration. (**B**) Changes in mouse body weight (*n* = 6). (**C**) Rate of weight change (*n* = 6). (**D**) Histopathological changes of spleen tissues by H&E staining in each group, magnification 20× (*n* = 3). Following exposure to hypoxic conditions, splenic tissue exhibited abnormalities, such as a decrease in lymphocyte numbers (black arrow). In addition, an increased number of neutrophils in the spleen tissue showed inflammatory infiltration (blue arrow) (The high-resolution images are available in Appendix A). (**E**) Spleen index (*n* = 6). (**F**) Quantity of white pulp in the spleen (*n* = 3). (**G**) Average area of white pulp in the mouse (*n* = 3). (**H**–**J**). Changes of main indexes of blood routine tests (*n* = 4–6). * *p* < 0.05, ** *p* < 0.01, *** *p* < 0.001, **** *p* < 0.0001 vs. NC.

**Figure 6 ijms-26-02584-f006:**
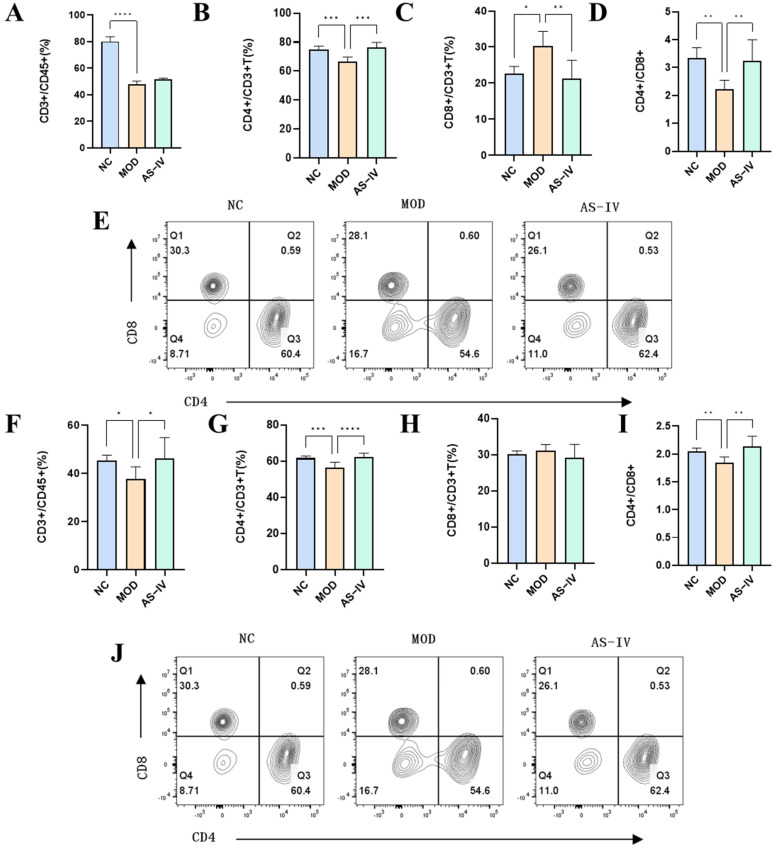
Effects of AS-IV on peripheral blood and spleen immune cells in mice with high altitude immune injury. (**A**) CD3^+^ T-lymphocyte in the blood (*n* = 6). (**B**) CD4^+^ T-lymphocyte in the blood (*n* = 6). (**C**) CD8^+^ T-lymphocyte in the blood (*n* = 6). (**D**) CD4^+^/CD8^+^ ratio in the blood (*n* = 6). (**E**) The proportions of CD4^+^ T-lymphocyte and CD8^+^ T-lymphocyte were detected by flow cytometry in the blood. (**F**) CD3^+^ T-lymphocyte in the spleen (*n* = 6). (**G**) CD4^+^T-lymphocyte in the spleen (*n* = 6). (**H**). CD8^+^ T-lymphocyte in the spleen (*n* = 6). (**I**). CD4^+^/CD8^+^ ratio in the spleen (*n* = 6). (**J**) The proportions of CD4^+^ T-lymphocyte and CD8^+^ T-lymphocyte were detected by flow cytometry in the spleen. * *p* < 0.05, ** *p* < 0.01, *** *p* < 0.001, **** *p* < 0.0001 vs. NC.

**Figure 7 ijms-26-02584-f007:**
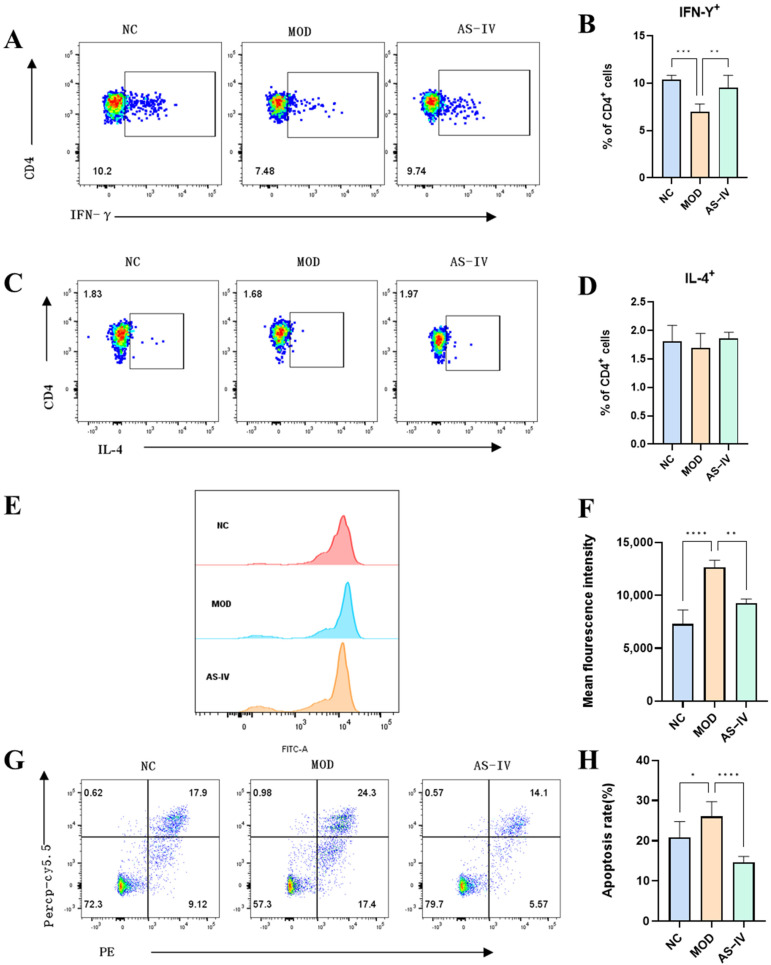
Effects of AS-IV on reactive oxygen species, apoptosis, and Th1/Th2 immune cells in the spleen of mice with high altitude immune injury. (**A**) Representative scatter plots and the ratio of the fraction of CD4^+^ IFN-γ^+^ cells. (**B**) CD4^+^ IFN-γ^+^ cell bar graphs (*n* = 5). (**C**) Representative scatter plots and ration of the fraction of CD4^+^ IL-4^+^ cells. (**D**) CD4^+^ IL-4^+^ cell bar graphs (*n* = 5). (**E**) Cell ROS scatter plot of spleen cells by flow cytometry. (**F**) Cell ROS bar graphs of spleen cells by flow cytometry (*n* = 5). (**G**) Cell apoptosis scatter plot of spleen cells by flow cytometry. (**H**). Cell apoptosis bar graphs of spleen cells by flow cytometry (*n* = 5). * *p* < 0.05, ** *p* < 0.01, *** *p* < 0.001, **** *p* < 0.0001 vs. NC.

**Figure 8 ijms-26-02584-f008:**
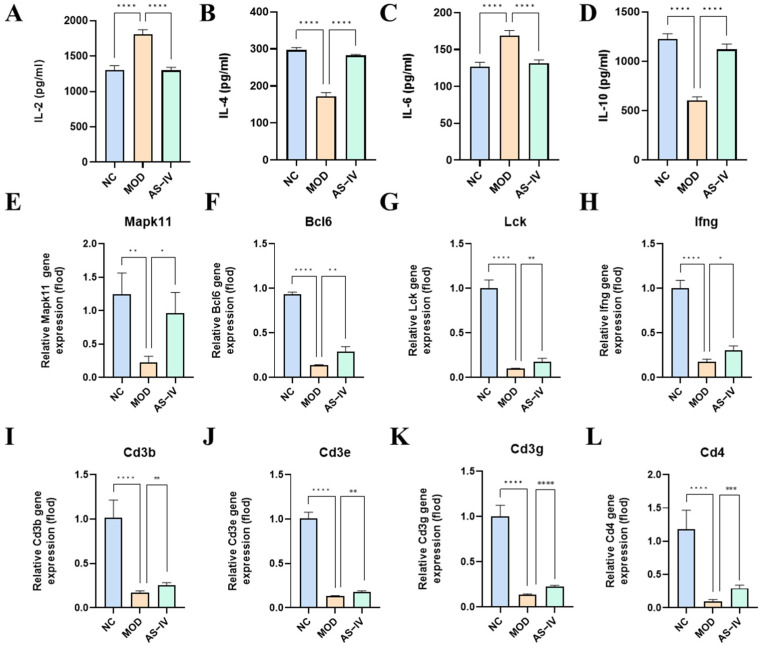
Effects of AS-IV on peripheral blood inflammatory factors and spleen mRNA in mice with high altitude immune injury. (**A**) Cytokine concentrations of IL-2 (*n* = 4). (**B**) Cytokine concentrations of IL-6 (*n* = 4). (**C**) Cytokine concentrations of IL-10 (*n* = 4). (**D**) Cytokine concentrations of IL-17A (*n* = 4). (**E**–**L**). AS-IV influenced the mRNA of master transcription factors and cytokine concentrations for Th1/Th2 (*n* = 3). * *p* < 0.05, ** *p* < 0.01, *** *p* < 0.001, **** *p* < 0.0001 vs. NC.

**Table 1 ijms-26-02584-t001:** Sequences of primers used in qRT-PCR.

Gene	5′-3′	3′-5′
*Mapk11*	GCGGGATTCTACCGGCAAG	GAGCAGACTGAGCCGTAGG
*Bcl6*	TAGAGCCCATAAGACAGTGCT	CACCGCCATGATATTGCCTTC
*Lck*	TGGAGAACATTGACGTGTGTG	ATCCCTCATAGGTGACCAGTG
*Ifng*	GCAACAGCAAGGCGAAAAAG	CGCTTCCTGAGGCTGGATTC
*Cd3d*	AGCGGGATTCTGGCTAGTCT	CGCTGGTATTGCAGGTCACAA
*Cd3e*	ATGCGGTGGAACACTTTCTGG	GCACGTCAACTCTACACTGGT
*Cd3g*	ACTGTAGCCCAGACAAATAAAGC	TGCCCAGATTCCATGTGTTTT
*Actb*	GGCTGTATTCCCCTCCATCG	CCAGTTGGTAACAATGCCATGT

## Data Availability

Data will be made available on request.

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
