# Peer review of "Th1/Th2 Immune Imbalance in the Spleen of Mice Induced by Hypobaric Hypoxia Stimulation and Therapeutic Intervention of Astragaloside IV"

_ijms, 2025, doi:10.3390/ijms26062584_

Round 1
Reviewer 1 Report
Comments and Suggestions for Authors
The present study aims to establish a hypobaric hypoxia immune injury model and explore the intervention and therapeutic effects of Astragaloside IV (AS-IV). This study first focuses on the dynamic changes in immune cells in hypobaric hypoxia environments to understand the complex physiological process and explore whether AS-IV possess the potential therapeutic effects on it. Despite the utilization of various methodologies in this study, the hypothesis was not adequately substantiated within this manuscript. This is because it cannot be conclusively demonstrated that the inflammatory factors in the blood and the master transcription factors associated with Th1/Th2 cells, as well as the genes involved in the Th1/Th2 signaling pathway, were exclusively derived from Th1/Th2 cells. It is recommended to investigate the expression of these inflammatory factors and related genes specifically within Th1/Th2 cells.
Several issues should be answered/improved from the presented manuscript:
1、Figure 1A and Figure 5A in this manuscript lack a clear description, making it difficult for the author to determine the precise timing of hypobaric hypoxia exposure and oral gavage. It is recommended that the initiative experiment time point commence from day 0 in the diagram.
2、It was reported that on day 8, the MOD and AS-IV groups were subjected to a hypobaric hypoxia chamber. However, the methodology for administering oral gavage under hypobaric hypoxic conditions post-day 8 was not clearly specified.
3、As is well known, lymphocytes do not include monocytes and macrophages. Monocytes are predominantly found in the blood, while macrophages are more abundant in the spleen. However, the methods described in sections 4.6 and 4.7 were deemed inaccurate as they failed to exclude macrophages and monocytes from both the spleen and blood samples, as required by the protocols outlined in this manuscript.
4、Lack of the method description of statistics of white pulp quantity in the spleen.
5、It appears that Figures 7E and 7G depict the detection of total reactive oxygen species (ROS) and apoptosis in splenic cells, rather than specifically in Th1 and Th2 cells. However, the title of Figure 7 is inaccurate as it describes the effects of AS-IV on ROS levels and apoptosis in Th1/Th2 immune cells and CD4+ T cells in the spleen of mice with high-altitude immune injury.
6、The legend for Figure 8 lacks an I-L description.
7、The sentence in lane 361 was incomplete: " To further verify the effect of plateau hypoxia on T cells, the mRNA levels of Makp11, Bcl6, Lck, Cd3b, Cd3e, Cd3g, Cd4, Runx1, Runx3, Il12rb2, and H2-Ab1 in the splenic Th1/Th2 signaling pathway were detected by." Moreover, the results were obtained from the spleen rather than from T cells. It is recommended to specifically examine the gene expression in Th1/Th2 cells instead of analyzing the entire spleen.
8、Among the listed genes (Makp11, Bcl6, Lck, Cd3b, Cd3e, Cd3g, Cd4, Runx1, Runx3, Il12rb2, and H2-Ab1), which are master transcription factors for Th1/Th2 differentiation, and which belong to the Th1/Th2 signaling pathway? This query arises from the observation in Figure 8 E-H that AS-IV influenced the mRNA expression of master transcription factors.
9、The discussion section is excessively lengthy and it is recommended to divide it into appropriately sized segments.
Comments on the Quality of English LanguageSeveral sentences in this manuscript were incomplete (e.g., line 361), and others were expressed unclearly (e.g., line 296). Additionally, the overall English requires further refinement.
Author Response
Comments 1: Figure 1A and Figure 5A in this manuscript lack a clear description, making it difficult for the author to determine the precise timing of hypobaric hypoxia exposure and oral gavage. It is recommended that the initiative experiment time point commence from day 0 in the diagram.
Response 1: Thank you for your suggestion. The initial experimental time point in Figure 1A and 5A has been started from day 0 in the figure according to your suggestion. The changes are in line 120 on page 3 and line 277 on page 9 of the text.
Comments 2: It was reported that on day 8, the MOD and AS-IV groups were subjected to a hypobaric hypoxia chamber. However, the methodology for administering oral gavage under hypobaric hypoxic conditions post-day 8 was not clearly specified.
Response 2: Thank you for your suggestion. As our administration method is preventive administration plus therapeutic administration, preventive administration is given for 7 days first, then therapeutic administration is given for 5 days, and continuous administration is given for 12 days. The administration method is the same every day, but the dosage needs to be reduced after the 8th day, It has been modified according to your request. The changes are listed on page 16, lines 495-506.
Modify the content “Eighteen male BALB/c mice were randomly divided into 3 groups: normoxia control group (NC group), model group (MOD group) and astragaloside ⅳ group (AS-IV group), with 6 mice in each group. AS-IV group: mice were gavaged with 80 mg/kg AS-IV daily for 12 days. NC group and MOD group were given equal volume of sodium carboxymethyl cellulose by gavage every day for 12 consecutive days. Each group was given prophylaxis on days 0-7. From day 8, the mouse in the MOD and AS-IV groups were placed in a hypo-baric hypoxia chamber and exposed to a simulated hypoxia environment at an altitude of 7000 m (308 mmHg, equivalent to Pio2 8.0 kPa) at a rate of 10 m/s, Treatment was per-formed by intragastric administration after daily destocking. Light, temperature, and hu-midity conditions were similar in the NC group, but normal atmospheric pressure and normoxia conditions were maintained in a separate room. Sampling and testing were performed on day 12.”
Comments 3: As is well known, lymphocytes do not include monocytes and macrophages. Monocytes are predominantly found in the blood, while macrophages are more abundant in the spleen. However, the methods described in sections 4.6 and 4.7 were deemed inaccurate as they failed to exclude macrophages and monocytes from both the spleen and blood samples, as required by the protocols outlined in this manuscript.
Response 3: Thank you for your suggestion. I'm sorry that the title of the previous experimental scheme was improperly expressed. 4.6 and 4.7 were flow pretreatment, and the cell types were mainly distinguished by antibody staining in the later period.
Comments 4: Lack of the method description of statistics of white pulp quantity in the spleen.
Response 4: Thank you for your suggestion. I have added the method of counting the number of white pulp in the spleen in lines 584 to 585 on page 17 of the article.
Modify the content ”The number and area of white pulp were measured with image analysis software, such as ImageJ, and statistical analysis was performed.”
Comments 5: It appears that Figures 7E and 7G depict the detection of total reactive oxygen species (ROS) and apoptosis in splenic cells, rather than specifically in Th1 and Th2 cells. However, the title of Figure 7 is inaccurate as it describes the effects of AS-IV on ROS levels and apoptosis in Th1/Th2 immune cells and CD4+ T cells in the spleen of mice with high-altitude immune injury.
Response 5: Thank you for your suggestion. The title and note of Figure 7 have been modified according to your requirements. The changes are listed on page 11, lines 318-325.
Modify the content “Figure 7. Fig. Effects of AS-IV on reactive oxygen species, apoptosis and Th1/Th2 immune cells in spleen of mice with high altitude immune injury. (A) Cell ROS scatter plot of spleen cells by flow cytometry; (B) Cell ROS Bar graphs of spleen cells by flow cy-tometry (n=5). (C) cell apoptosis scatter plot of spleen cells by flow cytometry; (D). cell apoptosis bar graphs of spleen cells by flow cytometry (n=5); (E) Representative scatter plots and ration of the fraction of CD4+IFN-γ+cells; (F) CD4+IFN-γ+ cell Bar graphs (n=5); (G) Representative scatter plots and ration of the fraction of CD4+IL-4+cells; (H) CD4+IL-4+ cell Bar graphs (n=5); *P < 0.05, **P < 0.01, ***P < 0.001, ****P < 0.0001 vs. NC.”
Comments 6: The legend for Figure 8 lacks an I-L description.
Response 6: Thanks for your suggestion, the description of I-L in Figure 8 has been added as required. The change is in line 354 on page 12 of the text.
Modify the content “(E)-(L). AS-IV influenced the mRNA of master transcription factors and Cytokine con-centrations for Th1 /Th2 (n=3).”
Comments 7: The sentence in lane 361 was incomplete: " To further verify the effect of plateau hypoxia on T cells, the mRNA levels of Makp11, Bcl6, Lck, Cd3b, Cd3e, Cd3g, Cd4, Runx1, Runx3, Il12rb2, and H2-Ab1 in the splenic Th1/Th2 signaling pathway were detected by." Moreover, the results were obtained from the spleen rather than from T cells. It is recommended to specifically examine the gene expression in Th1/Th2 cells instead of analyzing the entire spleen.
Response 7: Thank you for your suggestion. Since we detected the transcription of spleen during the previous modeling, and found the changes of Th1/Th2 cell pathway, we can further verify the expression of its gene by detecting the mRNA of spleen after administration. And because Th1/Th2 cells were not sorted out in the previous experiments, it was difficult to detect their gene expression.
Through literature research, it was also found that the changes in Th1/Th2 cell pathway were reflected by detecting the expression of related genes in Th1/Th2 cell pathway in mouse spleen tissue 1,2. The changes of Th1/Th2 cell pathway were verified by detecting the expression of related genes in the Th1/Th2 cell pathway in the small intestine of mice3.
(1) Xu, H.; Duan, X.; Wang, Y.; Geng, W. Amelioration Effect of Lactobacillus Kefiranofaciens ZW3 on Ovalbumin-Induced Allergic Symptoms in BALB/c Mice. Foods 2024, 14 (1), 16. https://doi.org/10.3390/foods14010016.
(2) Zhu, D.; Du, Y.; Zhao, M.; Ablikim, D.; Huang, H.; Pan, W.; Zeng, X.; Xu, C.; Lu, M.; Sutter, K.; Dittmer, U.; Zheng, X.; Yang, D.; Liu, J. Functional B Cell Deficiency Promotes Intrahepatic HBV Replication and Impairs the Development of Anti-HBV T Cell Responses. Hepatol Int 2024. https://doi.org/10.1007/s12072-024-10753-8.
(3) Tang, J.; Hu, Y.; Fang, J.; Zhu, W.; Xu, W.; Yu, D.; Zheng, Z.; Zhou, Q.; Fu, H.; Zhang, W. Huanglian Ejiao Decoction Alleviates Ulcerative Colitis in Mice Through Regulating the Gut Microbiota and Inhibiting the Ratio of Th1 and Th2 Cells. Drug Des Devel Ther 2025, 19, 303–324. https://doi.org/10.2147/DDDT.S468608.
Comments 8: Among the listed genes (Makp11, Bcl6, Lck, Cd3b, Cd3e, Cd3g, Cd4, Runx1, Runx3, Il12rb2, and H2-Ab1), which are master transcription factors for Th1/Th2 differentiation, and which belong to the Th1/Th2 signaling pathway? This query arises from the observation in Figure 8 E-H that AS-IV influenced the mRNA expression of master transcription factors.
Response 8: Thank you for your suggestion, major transcription factors for Th1/Th2 differentiation: Bcl6, Runx3. Th1/Th2 signaling pathway related genes: Ifng, Il12a, Il12rb2, Il4ra, Cd4, Cd247, Cd3b, Cd3e, Cd3g. Other indirectly affected Th1/Th2 signaling pathway related genes: H2-Ab1, H2-Aa, H2-Oa, H2-Ob, Lck, Runx1, Makp11.
Comments 9: The discussion section is excessively lengthy and it is recommended to divide it into appropriately sized segments.
Response 9: Thank you for your suggestions to make appropriate changes to the discussion section. The changes are on pages 13-15, lines 376-474.
Reviewer 2 Report
Comments and Suggestions for Authors
Dear Authors,
Thank You for the wonderful opportunity to review this study.
I have no questions or clarifications.
This paper serves as an exceptional model of research, demonstrating excellence in methodology, clarity, result interpretation, discussion, and, importantly, its perspectives and practical applications.
I only suggest expanding the conclusion section to further discuss the benefits of AS-IV pretreatment in modulating plateau immunoinjury by inhibiting inflammatory responses. Also, a minor correction is needed in the first line of the conclusion section.
Author Response
Comments 1:I only suggest expanding the conclusion section to further discuss the benefits of AS-IV pretreatment in modulating plateau immunoinjury by inhibiting inflammatory responses. Also, a minor correction is needed in the first line of the conclusion section.
Response 1: Thank you for your suggestion that the benefits of AS-IV pretreatment to modulate platform immune injury by inhibiting the inflammatory response have been further extended, and the modifications are in lines 376-474 on pages 13-15. In addition, the error in the discussion section has been modified, and the modified content is on page 18, line 635.
Round 2
Reviewer 1 Report
Comments and Suggestions for Authors
The authors addressed the concerns point by point on the web. However, some issues remain unresolved, particularly regarding question 7. They explained that "due to the fact that Th1/Th2 cells were not sorted out in the previous experiments, it was challenging to detect their gene expression." Although they did not conduct additional experiments, they provided several similar articles to demonstrate the desirability of their approach.
More importantly, the authors claimed to have revised the manuscript. However, since they did not highlight the revised portions, I did not observe any corresponding revisions in the file ijms-3447075-peer-review-v2. Further review is necessary before publication.
Comments on the Quality of English LanguageThe English expression in the file ijms-3447075-peer-review-v2 was identical to that of v1.
Author Response
Comments 1:
The authors addressed the concerns point by point on the web. However, some issues remain unresolved, particularly regarding question 7. They explained that "due to the fact that Th1/Th2 cells were not sorted out in the previous experiments, it was challenging to detect their gene expression." Although they did not conduct additional experiments, they provided several similar articles to demonstrate the desirability of their approach.
More importantly, the authors claimed to have revised the manuscript. However, since they did not highlight the revised portions, I did not observe any corresponding revisions in the file ijms-3447075-peer-review-v2. Further review is necessary before publication.
Response 1:
Thank you for your suggestion. We have modified the text according to your suggestion on the previous question 7, marked it with the blue highlighted highlight logo, in line 468-475 on page 15 of the text, and the reference is in line 734-738 on page 21.
I'm sorry about the " However, since they did not highlight the revised portions, I did not observe any corresponding revisions in the file ijms-3447075-peer-review-v2. "Probably due to a file upload error, the first revision was in purple font, This time, it's a blue highlight. Upload the name of the manuscript: ijms-3447075-2.18.
The specific changes are as follows:
In the this study, the spleen transcription in the high altitude immune injury model was detected, and the changes of Th1/TH2 cell differentiation pathway were found. Literature studies found that the mRNA levels of Th1/Th2 signaling pathway-related genes in the spleen were detected by PCR, and it was found that the changes of Th1/Th2 signaling in the spleen could also reflect the disorder of spleen immune function(36,37). The results of this study also showed that AS-IV treatment could improve the abnormality of Th1/Th2 cell pathway in the spleen of mice induced by hypobaric hypoxia.
- Zhou S-F, Ma J, Qu H-T, et al.: Characterization of Th1- and Th2-associated Chemokine Receptor Expression in Spleens of Patients with Immune Thrombocytopenia. J Clin Immunol 33: 938–946, 2013.
- Wang X, Ali W, Zhang K, et al.: The attenuating effects of serine against cadmium induced immunotoxicity through regu-lating M1/M2 and Th1/Th2 balance in spleen of C57BL/6 mice. Ecotoxicology and Environmental Safety 286: 117216, 2024.
